# Effect of CB1 Receptor Deficiency on Mitochondrial Quality Control Pathways in Gastrocnemius Muscle

**DOI:** 10.3390/biology13020116

**Published:** 2024-02-11

**Authors:** Rosalba Senese, Giuseppe Petito, Elena Silvestri, Maria Ventriglia, Nicola Mosca, Nicoletta Potenza, Aniello Russo, Francesco Manfrevola, Gilda Cobellis, Teresa Chioccarelli, Veronica Porreca, Vincenza Grazia Mele, Rosanna Chianese, Pieter de Lange, Giulia Ricci, Federica Cioffi, Antonia Lanni

**Affiliations:** 1Department of Environmental, Biological and Pharmaceutical Sciences and Technologies, University of Campania “Luigi Vanvitelli”, 81100 Caserta, Italy; rosalba.senese@unicampania.it (R.S.); giuseppe.petito@unicampania.it (G.P.); maria.ventriglia@hotmail.it (M.V.); nicola.mosca@unicampania.it (N.M.); nicoletta.potenza@unicampania.it (N.P.); aniello.russo@unicampania.it (A.R.); pieter.delange@unicampania.it (P.d.L.); 2Department of Sciences and Technologies, University of Sannio, 82100 Benevento, Italy; silves@unisannio.it; 3Department of Experimental Medicine, University of Campania “Luigi Vanvitelli”, 80138 Naples, Italy; francesco.manfrevola@unicampania.it (F.M.); gilda.cobellis@unicampania.it (G.C.); teresa.chioccarelli@unicampania.it (T.C.); veronica.porreca@unicampania.it (V.P.); vincenzagrazia.mele@unicampania.it (V.G.M.); rosanna.chianese@unicampania.it (R.C.); giulia.ricci@unicampania.it (G.R.)

**Keywords:** cannabinoid type 1 receptor (CB1), skeletal muscle, mitochondria, mitochondrial quality control (MQC), miRNA, mitochondrial unfolded protein response (UPRmt)

## Abstract

**Simple Summary:**

The cannabinoid type 1 receptor (CB1) is a primary cannabinoid receptor prominently found in the central nervous system and peripheral tissues, notably skeletal muscle. Muscle tissue CB1 is located on the cell and mitochondrial membrane. This study investigates how this receptor contributes to mitochondrial homeostasis in gastrocnemius muscle. The results indicate that CB1 absence induces a shift from fast-twitch to slow-twitch fibers, coupled with increased oxidative capacity and alterations in the antioxidant defense systems. Analysis of mitochondrial quality control (MQC) shows enhanced biogenesis, fusion, mitophagy, and UPR^mt^. Our study reveals the multifaceted impact of CB1 absence on the mitochondrial homeostasis of gastrocnemius muscle.

**Abstract:**

This study aims to explore the complex role of cannabinoid type 1 receptor (CB1) signaling in the gastrocnemius muscle, assessing physiological processes in both CB1^+/+^ and CB1^−/−^ mice. The primary focus is to enhance our understanding of how CB1 contributes to mitochondrial homeostasis. At the tissue level, CB1^−/−^ mice exhibit a substantial miRNA-related alteration in muscle fiber composition, characterized by an enrichment of oxidative fibers. CB1 absence induces a significant increase in the oxidative capacity of muscle, supported by elevated in-gel activity of Complex I and Complex IV of the mitochondrial respiratory chain. The increased oxidative capacity is associated with elevated oxidative stress and impaired antioxidant defense systems. Analysis of mitochondrial biogenesis markers indicates an enhanced capacity for new mitochondria production in CB1^−/−^ mice, possibly adapting to altered muscle fiber composition. Changes in mitochondrial dynamics, mitophagy response, and unfolded protein response (UPR) pathways reveal a dynamic interplay in response to CB1 absence. The interconnected mitochondrial network, influenced by increased fusion and mitochondrial UPR components, underlines the dual role of CB1 in regulating both protein quality control and the generation of new mitochondria. These findings deepen our comprehension of the CB1 impact on muscle physiology, oxidative stress, and MQC processes, highlighting cellular adaptability to CB1^−/−^. This study paves the way for further exploration of intricate signaling cascades and cross-talk between cellular compartments in the context of CB1 and mitochondrial homeostasis.

## 1. Introduction

Endocannabinoids are naturally occurring compounds in the body that interact with the endocannabinoid system, a complex network of receptors and enzymes [1,2,3]. The cannabinoid type 1 receptor (CB1) is a primary cannabinoid receptor within this system, prominently found in the central nervous system and peripheral tissues, notably skeletal muscle [4,5,6,7,8,9]. In muscle tissue, CB1s are located on the cell membrane, referred to as peripheral CB1 (pCB1) [4,10,11]. Interestingly, CB1 has also been identified within mitochondria, known as mitochondrial CB1 (mtCB1) [12,13,14,15,16].

The activation of CB1 has been associated with the regulation of various physiological processes, including food intake, energy expenditure, glucose metabolism, and lipid metabolism [5,16,17,18,19,20]. Nevertheless, research on the role of CB1 in modulating the metabolic and functional aspects of skeletal muscle is still ongoing.

Skeletal muscle, the most substantial tissue type in the human body, serves crucial roles essential for overall health [21]. This muscle is not only fundamental for maintaining posture and enabling mobility but also for regulating whole-body glucose homeostasis and thermogenesis. Comprising about 40% to 50% of total body mass in humans, skeletal muscle accounts for approximately 30% of basal energy expenditure [22].

It is well known that skeletal muscle metabolism is associated with fiber types. Type I fibers (slow twitch), rich in mitochondria, primarily utilize fatty acid oxidation to generate adenosine triphosphate (ATP). In contrast, type II fibers (fast twitch) prefer glucose as their energy source. These fibers exhibit remarkable flexibility, as they can adjust their traits in response to varying functional and energetic requirements.

Iannotti et al. demonstrated that the endocannabinoid 2-arachidonoylglycerol (2-AG), which acts via the CB1, inhibits the differentiation of primary human satellite cells [23]. More recently, Gonzalez-Mariscal et al. showed that skeletal muscle CB1 ablation increases muscle mass and improves physical performance, as well as enriching type 1 fibers [24].

Immunohistochemistry data from the laboratory of Koppo and colleagues showed that CB1 was higher in slow human compared to fast human muscle fibers, indicating that CB1 could play a role in influencing muscle fiber type [8]. However, more studies are required to understand the effect of CB1 on fiber-type switching in skeletal muscle. It is also unclear whether microRNAs (miRNAs), key regulators of muscle energy metabolism and fiber type, are involved in CB1-induced transcriptional circuits controlling skeletal muscle gene expression. Muscle fibers’ ability to adapt to different metabolic needs depends largely on the regulation of mitochondrial function. In skeletal muscle, CB1s are predominantly localized in the mitochondria, indicating a direct influence on mitochondrial function [25].

Mitochondria, key organelles within skeletal muscle cells, play an essential role in several vital processes. Renowned as the powerhouses of cells, they participate in managing energy-related signaling pathways, generating and regulating reactive oxygen species (ROS), maintaining calcium equilibrium, and controlling apoptosis [26]. Given their critical role in maintaining skeletal muscle functionality and adaptability, mitochondrial impairments are linked to various skeletal muscle-related health issues [27]. The presence of mtCB1, in addition to that of pCB1, is particularly intriguing, as it suggests a direct role of CB1 in modulating mitochondrial activities [25].

The interaction of CB1 with mitochondria in skeletal muscle cells could influence cellular energy efficiency and metabolic flexibility, affecting overall muscle function and adaptation to various physiological demands. The dynamic balance of mitochondrial biogenesis, fusion, fission, and mitophagy, collectively referred to as mitochondrial quality control (MQC), is essential for maintaining muscle health [28]. MQC, through the coordination of the abovementioned processes, ensures cell and mitochondrial homeostasis. Alteration in any of these pathways compromises mitochondrial quality and may potentially lead to impaired muscle function. Another important mechanism involved in MQC is the mitochondrial unfolded protein response (UPR^mt^), which plays a pivotal role in maintaining mitochondrial proteostasis [29].

The UPR^mt^ is activated in response to the accumulation of unfolded or misfolded proteins within the mitochondrial matrix, a condition that can arise from oxidative stress, environmental challenges, or genetic mutations affecting mitochondrial function. When activated, the UPR^mt^ initiates a signaling cascade that enhances the expression of mitochondrial chaperones and proteases, aiming to restore protein folding homeostasis. Additionally, it can modulate the expression of antioxidant response genes and influence metabolic reprogramming to adapt to cellular stress, underscoring its broader role in cellular defense mechanisms and the maintenance of mitochondrial and cellular function. Several studies have demonstrated that UPR^mt^ and endoplasmic reticulum UPR (UPR^ER^) have intersecting crossroads as a result of disruption of homeostasis. In some conditions, these crossroads can integrate their signaling and co-activate, thereby enhancing ER and mitochondrial function when proteostasis is dysregulated [30,31,32]. It has been suggested that PKR-like endoplasmic reticulum kinase (PERK) signaling pathways coordinate UPR^mt^ and UPR^ER^ [33,34]. The PERK protein is located at the ER-mitochondria contact site, providing a PERK-mitochondria signaling pathway that detects stress arising from the proteostasis of both organelles. PERK/ATF4 signaling, which is crucial for UPR^ER^ [30], also contributes to UPRmt [32]. By activating PERK/eIF2 signaling, mitochondrial complexes produce fewer reactive oxygen species. These responses contribute to overall cellular health and play critical roles in adapting to stress, preventing the accumulation of dysfunctional proteins, and promoting cell survival. Dysregulation of these responses has been implicated in various diseases, including neurodegenerative disorders and metabolic diseases, highlighting their importance in maintaining cellular and organismal homeostasis. 

This study aims to understand the role of CB1 signaling in remodeling contractile myofiber composition and in regulating the mitochondrial compartment of gastrocnemius muscle to gain a comprehensive overview. It focuses on examining the effects of CB1 absence on muscle fiber shift, oxidative capacity, and various crucial mitochondrial parameters in the gastrocnemius muscle, including cytochrome oxidase activity, functional organization of the respiratory chain, levels of oxidative stress, antioxidant defense mechanisms, and various aspects of MQC, such as mitochondrial dynamics (fusion and fission), mitophagy, biogenesis, DNA repair, and the UPR^mt^.

## 2. Materials and Methods

### 2.1. Animals and Animal Care

Mice (Mus musculus) genetically deleted for CB1 were provided by Prof. Ledent [35]. Male and female CB1 heterozygous (CB1^+/−^) mice have been maintained on a CD1 background (Charles River Laboratories, Lecco, Italy), then used to expand the colony and generate adult CB1^+/+^, CB1^−/−^, and CB1^+/−^ male mice. The animals were maintained in a temperature-controlled room at 22 °C under a 12 h dark/light cycle and fed a pellet diet with free access to water. Adult males (3–5 months) under anesthesia were sacrificed for cervical dislocation. For later processing, the skeletal muscle (gastrocnemius) was excised, weighed, and frozen in liquid nitrogen. Gastrocnemius muscle was dissected as follows: the skin was cut longitudinally towards the legs without touching the muscles and tendons. Subsequently, we used a pin to attach the leg to the support at foot level. We began isolating the skeletal muscles as follows: anterior tibial muscle, extensor digitorum longus, gastrocnemius, and soleus. Gastrocnemius will become separate from other muscles as soon as you lift the Achilles tendon. All animals received human care according to the criteria outlined in the Guide for the Care and Use of Laboratory Animals, prepared by the National Academy of Sciences and published by the National Institutes of Health. To minimize suffering and pain for animals, every effort was made. The sample size (*n* = 5) that allowed us to obtain statistically significant results was calculated using a G* Power Test, developed by the University of Dusseldorf (http://www.gpower.hhu.de/ (accessed on 22 January 2022). G Power can calculate sample size based on pre-designed effect sizes at small, medium, and large differences between the groups based on Cohen’s principles [36]. Experiments involving animals were approved by the Italian Ministry of Education and the Italian Ministry of Health, with authorization n°941/2016-PR issued on 10 October 2016.

### 2.2. Transmission Electron Microscopy (TEM) Analysis

To evaluate skeletal muscle fine sub-cellular ultrastructural features, CB1^+/+^ and CB1^−/−^ skeletal muscle were minced in small pieces and fixed overnight (O.N.) in glutaraldehyde 2.5% (Electron Microscopy Science, Hatfield, PA, USA) in 0.1 M cacodylate buffer (pH 7.4) at 4 °C. Fixed samples were then rinsed with 0.1 M cacodylate buffer for at least 1 h, post-fixed with 1% osmium tetroxide (OsO4) in 0.1 M cacodylate buffer (Electron Microscopy Science, Hatfield, PA, USA), dehydrated in ethanol (Sigma-Aldrich, Milano, Italy, EU), and embedded in epoxy resin (Electron Microscopy Science, Hatfield, PA, USA). Ultrathin sections (60 nm), obtained using an UC6 ultramicrotome (Leica, Wetzlar, Germany, EU) equipped with a diamond knife (DiATOME US, Hatfield, PA, USA), were placed on copper grids (Electron Microscopy Science, Hatfield, PA, USA). Ultrathin sections were then treated with Uranyl Acetate Replacement stain (UAR-Electron Microscopy Science, Hatfield, PA, USA) and contrasted with lead citrate (Sigma-Aldrich, Milano, Italy, EU). Samples were studied using a 100 kV transmission electron microscope EM208S PHILIPS (FEI—Thermo Fisher, Waltham, MA, USA) equipped with the acquisition system Megaview III SIS camera (Olympus/EMSIS) and iTEM3/Radius software, version 2.1.

### 2.3. Measurement of Hydrogen Peroxide (H_2_O_2_) in Skeletal Muscle Samples

H_2_O_2_ was measured in skeletal muscle samples using the Hydrogen Peroxide Assay Kit (Colorimetric/Fluorimetric) (Abcam, ab102500). Briefly, skeletal muscle tissues (40 mg) were homogenized rapidly in H_2_O_2_ Assay buffer (supplied from the kit) using an UltraTurrax homogenizer and then centrifuged at 10,000× *g* in a Beckman Optima TLX Ultracentrifuge (Beckman Coulter S.p.A., Milan, Italy) for 2–5 min at 4 °C. Subsequently, a deproteinization protocol was performed. For the deproteinization protocol, we used perchloric acid (PCA) at 4 M, and the excess PCA was precipitated by adding ice-cold 2 M KOH, which equals 34% of the supernatant. After neutralization, the pH was adjusted to 6.5–8 by adding 0.1 M KOH or PCA. The sample was centrifuged at 13,000× *g* for 15 min at 4 °C. Subsequently, we prepared a master mix of the reaction mix (Assay Buffer, OxiRed Probe, Developer Solution V/HRP) and added 50 µL of the Reaction Mix and 50 µL of the sample to each well. After incubation for 10 min at room temperature, fluorescence was measured using a microplate reader, BioTek Sinergy H1 (Ex/Em = 535/587 nm) (Agilent, Santa Clara, CA, USA).

### 2.4. Mitochondria Isolation

Differential centrifugation was used to isolate mitochondria from skeletal muscle. An isolation medium consisting of 220 mM mannitol, 70 mM sucrose, 20 mM TRIS-HCl, 1 mM EDTA, and 5 mM EGTA (pH 7.4) supplemented with 0.1% BSA was used to gently homogenize tissue fragments using a Potter-Elvehjem homogenizer (Heidolph Instruments, Schwabach, Germany). The homogenate was centrifuged at 800× *g* for 10 min at 4 °C. Subsequently, the supernatant was separated from the pellet and centrifuged at 3000× *g* for 30 min at 4 °C. For later use, mitochondrial pellets were washed twice, resuspended in an isolation medium, and kept on ice or at −80 °C.

### 2.5. Determination of Cytochrome Oxidase Activity in Skeletal Muscle Mitochondria

Aliquots of mitochondria were incubated for 30 min at 0 °C after the addition of 1.0 mg/mL lubrol. Cytochrome oxidase activity was determined polarographically at 37 °C by using the Oroboros 2k-Oxygraph system instrument (O2k, OROBOROS INSTRUMENTS, Innsbruck, Austria). Mitochondrial homogenates were incubated with 30 μM Cytochrome C, 4 μM Rotenone, 0.5 mM Dinitrophenol, 10 mM Na-malonate, and 75 mM HEPES at pH 7.4 in 2 mL of reaction medium. A substrate of 4 mM Na ascorbate with 0.3 mM of N,N,N′,N′-tetramethyl-p-phenylenediamine was added after 10 min to determine oxygen consumption. The auto-oxidation of the substrate was evaluated in parallel measurements in the absence of mitochondrial homogenate. Sample protein content was determined using Bio Rad’s DC method (Bio-Rad Laboratories, s.r.l., Segrate, Italy).

### 2.6. Separation of Respiratory Complexes and Supercomplexes by Blue-Native Page (BN-PAGE) and Histochemical Staining for In-Gel Activity

Solubilization of mitochondrial membranes by detergents, BN-PAGE, staining, and densitometric quantification of oxidative phosphorylation complexes was performed essentially as described by Schagger et al. [37] and Silvestri et al. [38] with minor modifications. Briefly, the mitochondria-containing sediment was suspended in a low salt buffer (50 mM NaCl, 50 mM imidazole, pH 7.0) and solubilized with 10% dodecyl-maltoside (for solubilization of individual respiratory chain complexes) or digitonin (4 g/g protein, for solubilization of respiratory chain supercomplexes). Immediately after the electrophoretic run (carried out on 6–13% (*w*/*v*) gradient polyacrylamide gels), enzymatic colorimetric reactions were performed essentially as reported by others [39]. The gel was run in a refrigerator or a cold room with a constant voltage of 100 V. When the samples had completely entered the gel, the current was changed to 15 mA until the dye front reached approximately one-third from the top of the gel. The electrophoresis was stopped when the dye front reached the bottom of the gel. Complex I activity was determined by incubating the gel slices with 2 mM Tris–HCl, pH 7.4, 0.1 mg/mL NADH, and 2.5 mg/mL nitro blue tetrazolium (NTB) at room temperature. Complex II activity was evaluated after incubating the gel slices in a 100 mM Tris/glycine buffer at pH 7.4 containing 1 mg/mL NTB and 1 mM sodium succinate. Complex IV activity was estimated by incubating BN-PAGE gels with 5 mg 3,3′-diaminobenzidine tetrahydrochloride (DAB) dissolved in 9 mL phosphate buffer (0.05 M, pH 7.4), 1 mL catalase (20 μg/mL), 10 mg cytochrome c, and 750 mg sucrose. The original color of the complex I, II, or IV-reacting bands was preserved by fixing the gels in 50% (*w*/*v*) methanol and 10% (*w*/*v*) acetic acid. In parallel, another electrophoretic run was performed to stain the gels with Coomassie Blue G and obtain the total band pattern of respiratory complexes or supercomplexes. After gel scanning, the areas of the bands were expressed as absolute values (arbitrary units). For densitometric analysis, electronic images of the gels were acquired by means of a GS-800 calibrated densitometer (Bio-Rad) and analyzed using QuantityOne software, version 2.1 (Bio-Rad). Scanned gel images were processed for the removal of background and automatic detection of bands.

### 2.7. Preparation of Mitochondrial Lysates from the Skeletal Muscle

The mitochondrial pellet was resuspended in RIPA buffer containing 150 mM NaCl, 1.0% Triton X-100, 0.5% sodium deoxycholate, 0.1% SDS, 50 mM Tris, pH 8.0, supplemented with 1 mM Na_3_VO_4_, 1 mM PMSF, and 1 mg/mL leupeptin. The homogenate was left on ice for 1 h and shaken every 10 min. The protein concentrations of the homogenates were determined as described in Section 2.5. 

### 2.8. Preparation of Total Lysates from the Skeletal Muscle

Skeletal muscle samples were homogenized in Lysis Buffer containing 20 mM Tris-HCl (pH 7.5), 150 mM NaCl, 1 mM EDTA, 1 mM EGTA, 2.5 mM Na_2_H_2_P_2_O_7_, 1 mM b-CH_3_H_7_O_6_PNa_2_, 1 mM Na_3_VO_4_, 1 mM PMSF, 1 mg/mL leupeptin, and 1% (*v*/*v*) Triton X-100 (Sigma-Aldrich, St. Louis, MO, USA) using an UltraTurrax homogenizer, and then centrifuged at 16,000× *g* in a Beckman Optima TLX Ultracentrifuge (Beckman Coulter S.p.A., Milan, Italy) for 15 min at 4 °C. The supernatants were ultra-centrifuged at 40,000× *g* in a Beckman Optima TLX ultracentrifuge for 20 min at 4 °C. The protein concentrations of the supernatants were determined by the method described previously.

### 2.9. Western Blot Analysis

Electrophoreses on SDS-PAGE gels and Western blot analysis were performed essentially as described by Petito et al. [40] and Senese et al. [41], with minor modifications. Lysates containing approximately 30 µg of protein for skeletal muscle were diluted in an equal volume of 5× Laemmli’s reducing sample buffer (62.5 mM Tris pH 6.8, 10% (*w*/*v*) glycerol, 2% (*w*/*v*) SDS, 2.5% (*w*/*v*) pyronin, and 200 mM dithiothreitol), incubated at 95 °C for 5 min, and loaded onto 8% (*w*/*v*), 10% (*w*/*v*), or 12% (*w*/*v*) acrylamide/bisacrylamide gels. The gels were run with a low voltage (60 V) for separating gel and a higher voltage (140 V) for stacking gel in a Tris-Glycine running buffer (25 mM Tris, 192 mM glycine, 0.1% (*w*/*v*) SDS, pH 8.3). The gel was run for approximately an hour, or until the dye front ran off the bottom of the gel. To confirm the size of each protein, they were run alongside molecular weight ladders using Precision Plus Protein™ All Blue Prestained Protein Standards (Biorad, Hercules, CA, USA). Upon electrophoresis on SDS-PAGE gels, the proteins were transferred to nitrocellulose membranes (Thermofisher Scientific, Waltham, MA, USA) in a transfer buffer (50 nm Tris-HCI (pH 8.0), 20% (*w*/*v*) methanol, and 0.1% (*w*/*v*) SDS).

The membranes were blocked with 5% (*w*/*v*) nonfat dry milk (in TBS-T). Subsequently, they were probed with the following antibodies: Total OXPHOS (Abcam-1:500 dilution), PGC1α (Millipore-1:1000 dilution), NRF1 (Abcam-1:20,000 dilution), TFAM (Santa Cruz Biotechnology-1:1000 dilution), OPA1 (Abcam-1:1000 dilution), Drp1 (Cell Signaling-1:1000 dilution), Mfn2 (Abcam-1:1000 dilution), APE-1 (Novus Biologicals-1:1000 dilution), OGG1 (Novus Biologicals-1:1000 dilution), POL-γ (Santa Cruz Biotechnology-1:1000 dilution), Parkin (Cell Signaling-1:1000 dilution), Pink1 (Abcam-1:1000 dilution), Ambra1 (Cell Signaling-1:1000 dilution), LC3B (Santa Cruz Biotechnology-1:1000 dilution), SQSTM1/p62 (Cell Signaling-1:1000 dilution), CATALASE (Merck-1:750 dilution), SOD-2 (Abcam-1:1000 dilution), GPX-1 (GeneTex-1:1000 dilution), P-PERK (Thr980) (Cell Signaling-1:1000 dilution), PERK (Cell Signaling-1:1000 dilution), P-eiF2α (Cell Signaling-1:1000 dilution), eiF2α (Cell Signaling-1:1000 dilution), LONP1 (ABclonal-1:1000 dilution), CLPP (ABclonal-1:1000 dilution), TRAP1 (ABclonal-1:1000 dilution), ATF5 (ABclonal-1:1000 dilution), ATF4 (ABclonal-1:1000 dilution), CHOP (ABclonal-1:1000 dilution), Myosin skeletal slow (NOQ7.5.4D) (GeneTex-1:1000 dilution), Skeletal myosin (FAST) (Sigma Aldrich-1:1000 dilution), VDAC1 (GeneTex-1:1000 dilution), and B-ACTIN (Bioss Antibodies-1:1000 dilution). All antibody brands and concentrations used in our experimental model are reported in the Appendix A. 

As secondary antibodies, peroxidase anti-rabbit IgG (abcam-1:4000 dilution) and peroxidase anti-mouse IgG (abcam-1:4000 dilution) were used. Chemiluminescent blots were imaged using the ChemiDoc XRS+ (Biorad) and analyzed using Image Lab Software 6.0.1 for densitometric analysis. 

A band’s density was determined by adjusting the precise width of the band by using the “Lane Profile” tool in order to account for the area below the shaded peak. Background subtraction was set using the rolling disc setting in the “Lanes” tool. 

### 2.10. miRNA Isolation and Real-Time PCR Analyses from Skeletal Muscle

Total RNA (including miRNA) was extracted by the miRNeasy mini kit (QIAGEN, Hilden, Germany) from skeletal muscle samples according to the manufacturer’s protocol. Retrotranscription and Real-Time PCR were performed essentially as described by Petito et al. [42], with minor modifications. Total RNA (1 μg) was used to synthesize cDNA strands in a 20 μL reaction volume using the SuperScript IV Reverse Transcriptase for RT-PCR (Invitrogen). Then, 50 μM of random hexamers, 10 mM of dNTP mix, and 1 μg of total RNA were combined and heated at 65 °C for 5 min and then incubated on ice for at least 1 min. Annealed RNA was combined with the RT reaction mix and incubated at 23 °C for 10 min, 50–55 °C for 10 min, and 80 °C for 10 min. Real-time quantitative RT-PCR (RT-qPCR) was conducted with 50 nM gene-specific primers, IQ SYBR Green supermix (Bio-Rad), and cDNA samples (2 μL) in a total volume of 25 μL. A melting curve analysis was completed following amplification from 55 °C to 95 °C to ensure product identification and homogeneity. The mRNA expression levels were repeated in triplicate and normalized to a reference gene (B-ACTIN and GAPDH, stable under specific experimental conditions) by using the 2^−ΔΔCT^ method. PCR primers (Table 1) were designed using the Primer 3 program and synthesized and verified by sequencing at Eurofins Genomics (Ebersberg, Germany) [43]. MiRNA (2 ng) was used to synthesize cDNA strands in a 20 μL reaction volume using the MultiScribe™ Reverse Transcriptase (Applied Biosystems, Waltham, MA, USA). miR-499 and miR-208b were detected and quantified by RT-qPCR with TaqMan^®^ miRNA assays from Applied Biosystems, in conformity with the manufacturer’s protocol. The expression levels of these miRNAs were normalized to a reference gene (RNU6B) by using the 2^−ΔΔCT^ method. The analyses were performed on five independent experiments (*n* = 5) each in triplicate. 

### 2.11. Statistical Analysis 

Comparisons were performed using GraphPad Prism 8.0.1. The values were compared by the Student’s *t*-test for between-group comparisons. Differences with *p* < 0.05 were considered statistically significant. All data were expressed as the mean ± SEM of at least 5 independent animals (*n* = 5).

## 3. Results

### 3.1. CB1 Deficiency Affects the Structure and Oxidative Capacity of Gastrocnemius Muscle in Mice

Because CB1 affects skeletal muscle fiber-type profiles, we first measured the levels of MHC isoforms IIb (Myofast) and Ib (Myoslow) in gastrocnemius. Western blot analysis revealed that the MHCIb level was significantly increased in gastrocnemius from CB1^−/−^ mice when compared with CB1^+/+^, indicating an ongoing structural shift toward the slow/oxidative phenotype (Figure 1A). The MHCIIb level, on the other hand, was significantly reduced in gastrocnemius muscles from CB1^−/−^ mice (Figure 1A). In addition, the expression levels of two myomiRs located in the intron of myosin genes, miR-208b and miR-499, were overexpressed in CB1^−/−^ mice when compared with CB1^+/+^ mice, suggesting that the observed shift in mice lacking CB1 could be related to these mytomiRs (Figure 1B). Subsequently, we evaluated the expression levels of the target genes of these miRNAs such as Transcription factor 12 (TCF12), Folliculin Interacting Protein 1 (FNIP1), SRY-Box Transcription Factor 6 (*sox6*), and Regulator of differentiation 1 (*rod1*). Our results demonstrated that, as expected, the expression of miRNAs target genes showed a reversed pattern compared to that of miR-208b and miR-499, respectively.

In order to get an idea of oxidative capacity, we then measured specific cytochrome C oxidase (COX) activity in the skeletal muscle mitochondria of CB1^+/+^ and CB1^−/−^ mice. This activity was significantly increased in the skeletal muscle of CB1^−/−^ mice (Figure 1C).

### 3.2. CB1 Deficiency Affects Respiratory Chain Functional/Structural Organization of Gastrocnemius Muscle

The protein levels of OXPHOS complexes in the mitochondria of CB1^−/−^ mice were unchanged when compared to mitochondria isolated from CB1^+/+^ mice (Figure 2A). This result was confirmed by BN-PAGE analysis of OXPHOS complexes obtained from the skeletal muscle mitochondria of CB1^+/+^ and CB1^−/−^ mice. Figure 2B displays the five major OXPHOS complexes (I–V) in Coomassie blue staining (Figure 2B). As far as their amount, densitometric analysis revealed that there were no significant differences between mitochondria from CB1^+/+^ and CB1^−/−^ mice (Figure 2D). The in-gel activities of purified OXPHOS complexes I, II, and IV were estimated by an examination of complex-specific enzymatic colorimetric reactions. The stained enzymatic activities of the assayed complexes were localized specifically to a single band (Figure 2C). CB1^−/−^ mice showed significantly increased in-gel activity of both complex I and complex IV (Figure 2E,F). The significant increase in the in-gel activity of complex I corresponded to a decrease in the in-gel activity of complex II, which, however, did not reach statistical significance (Figure 2G).

To elucidate whether the knockout of the CB1 gene also alters the functional/structural organization of the respiratory chain in terms of the assembly and activity of supercomplexes, we used the mild detergent digitonin to extract muscle mitochondria from gastrocnemius muscles, as it retains supercomplexes within the mitochondrial membranes, which were then resolved by BN-PAGE and analyzed for complex I- and complex IV-in-gel activity. Complex I activity resulted in being present in four high molecular mass supercomplexes (SCI2-5), within the mass range of 720–1236 KDa (Figure 3A–C), the highest activity corresponding to the band SCI5 (Figure 3G). As far as it concerns complex IV, the activity of such enzyme resulted in being present in five SCs (SCIV2, 5, 6, 8, and 11), within the mass range of 242–1236 KDa (Figure 3A–F), with the highest activity corresponding to the lowest molecular mass band, namely, SCIV11 (Figure 3H). In some heavier SCs, CI and CIV activity coexisted, specifically in SCI/IV2 and SCI/IV5. When comparing the BN-PAGE blue-colored supercomplex profiles between CB1^−/−^ and CB1^+/+^ mice, the densitometric analysis revealed that no significant differences were identified (Figure 3D). In CB1^−/−^ mice, a significant increase in the in-gel activity of CI was detected in the SC bands SCI2 and SCI5. In addition, a significant increase in the in-gel activity of CIV was detected in SCIV6, SCIV8, and SCIV11 (Figure 3G,H). What was observed for the in-gel activity of CI- and CIV-containing SCs paralleled what was observed for the individual respiratory complexes.

### 3.3. Mitochondrial Antioxidant Defense Is Affected by CB1 Deficiency in Gastrocnemius Muscle

ROS are produced in response to an increase in respiratory activity. A significant increase in mitochondrial hydrogen peroxide (H_2_O_2_) production was observed in the gastrocnemius muscle of CB1^−/−^ mice when compared with CB1^+/+^ mice (Figure 4A). CATALASE, superoxide dismutase 2 (SOD-2), and glutathione peroxidase 1 (GPX-1), all antioxidant enzymes, play an essential role in protecting cells against oxidative damage. CB1^−/−^ mice have significantly decreased SOD-2 and GPX-1 protein levels compared to CB1^+/+^ mice, as shown in Figure 4B. However, CB1^−/−^ animals showed a 35% increase in CATALASE protein levels when compared with CB1^+/+^ but did not reach statistical significance (Figure 4B). The observed differences between CB1^+/+^ and CB1^−/−^ mice in mitochondrial ROS production and antioxidant enzyme protein levels suggest oxidative damage in the gastrocnemius muscle of CB1^−/−^ animals as a result of increased oxidative capacity. 

### 3.4. MQC Processes Are Affected by CB1 Deficiency in Gastrocnemius Muscle

Oxidative stress may be intrinsically linked to mitochondrial dysfunction. In order to investigate mitochondrial features in the absence of CB1, key players in mitochondrial quality control mechanisms were evaluated. Changes in the protein levels of peroxisome proliferative-activated receptor gamma coactivator 1α (PGC1α), nuclear respiratory factor 1 (NRF1), and mitochondrial transcription factor A (TFAM), markers of mitochondrial biogenesis, were evaluated. Western blot analysis revealed that protein levels of NRF1 and TFAM were significantly increased in CB1^−/−^ mice vs. CB1^+/+^ mice (Figure 5A); the protein levels of PGC1α were increased by about 30% in CB1^−/−^ mice. Mitofusin 2 (Mfn2) and optic atrophy 1 (OPA1), which are located in the outer and inner mitochondrial membranes, respectively, are GTPase proteins that regulate mitochondrial fusion. On the other hand, dynamin-related protein 1 (Drp1) controls mitochondrial fission. To further understand how mitochondrial dynamics are affected by CB1^−/−^, we next examined the protein levels of these above-mentioned markers intrinsically associated with mitochondrial dynamics. As reported in Figure 5B, the protein levels of OPA1 were significantly increased in the skeletal muscle of CB1^−/−^ mice compared to the wild-type animals, while the protein levels of Mfn2 and Drp1 were increased by about 7% and 15%, respectively, but did not reach statistical significance (Figure 5B). In addition, the analysis of ultrathin sections allowed us to observe that intermyofibrillar mitochondria (red arrowhead) appeared more numerous and, frequently, larger in CB1^−/−^ derived samples than wild-type ones (Figure 5C–F). These data showed that mitochondrial fusion was increasing. The increase in fusion is also associated with an increase in mitophagy, another key component of mitochondrial quality maintenance. In mammalian cells, mitophagy is mediated by two key proteins: outer mitochondrial membrane kinase Pink1 (PTEN-induced kinase 1) and cytosolic Parkin (E3 ubiquitin ligase). As shown in Figure 5G, Parkin and Pink1 levels were significantly increased in CB1^−/−^ mice (Figure 5G). A measure of sequestosome 1 (SQSTM1/p62) and microtubule-associated protein 1 light chain 3 isoform B (LC3B), two markers of autophagy, was also carried out. SQSTM1/p62 elevated levels inhibit autophagy, whereas decreased levels activate it. Significantly increased protein levels of LC3B and significantly decreased protein levels of SQSTM1/p62 were observed in the skeletal muscle homogenates of CB1^−/−^ mice compared to wild-type animals (Figure 5G). Furthermore, the activating molecule in beclin 1-regulated autophagy 1 (Ambra1), which is a critical regulator of autophagy [44], was significantly increased in CB1^−/−^ mice (Figure 5G). These results indicate an increase in mitophagy. Increased markers of mitophagy and fusion are part of a more extensive cellular response to preserve mitochondrial integrity and function. In fact, these mechanisms are activated in stressful situations, such as oxidative damage caused by ROS production in the gastrocnemius of CB1^−/−^ mice. Base excision repair (BER) also impacts mitochondrial homeostasis. Since BER plays a crucial role in maintaining mtDNA integrity, we examined the expression levels of its components. Compared to CB1^+/+^ mice, CB1^−/−^ mice showed a significant increase in apurinic/apyrimidinic endonuclease 1 (APE-1) protein levels and a non-significant increase of approximately 15% in DNA polymerase (POL-γ) protein levels (Figure 5H). Moreover, 8-oxoguanine glycosylase 1 (OGG1) protein levels were unchanged in CB1^−/−^ mice vs. CB1^+/+^ animals (Figure 5H). These data suggest that CB1 deficiency increases mtDNA repair by enhancing part of the BER pathway.

In addition to mitochondrial biogenesis, dynamics, mitophagy, and mtDNA repair, UPR^mt^ also plays a role in mitochondrial homeostasis. A canonical UPR^mt^ axis was analyzed. CB1^−/−^ increases UPR^mt^. The protein levels of proteases Lon peptidase 1 (LONP1) and caseinolytic mitochondrial matrix peptidase proteolytic subunit (CLPP) increased significantly in CB1^−/−^ mice, as well as the protein levels of the chaperone Tumor necrosis factor receptor-associated protein 1 (TRAP1), transcriptional factors such as activating transcription factors 4 (ATF4), and CEBP homologous protein (CHOP) (Figure 6A). Instead, the protein levels of activating transcription factors 5 (ATF5) were unchanged (Figure 6A). Finally, since UPR^mt^ and UPR^ER^ have crossroads and PERK signaling pathways seem to be involved, we measured the phosphorylation levels of P-PERK and P-eiF2α. The phosphorylation of both of these proteins, indicating their activation, was significantly increased in CB1^−/−^ mice compared to wild-type mice (Figure 6B). This suggests that UPR^ER^ is also active in CB1-lacking mice. 

## 4. Discussion

This study investigates the complex role of CB1 in the gastrocnemius muscle, aiming to provide new insights into various physiological processes influenced by CB1 signaling. Our main focus is specifically on deepening our understanding of the involvement of CB1 in several aspects of mitochondrial homeostasis, a critical component of cellular function that holds broad implications for overall muscle physiology. Particular focus has been directed towards the relationship between CB1 function and UPR^mt^, as it remains an unexplored area of study thus far.

At tissue level, the results show a significant decrease in MyoFast (MHCIIb) protein levels and a simultaneous increase in MyoSlow (MHCIb) protein levels in the gastrocnemius of CB1^−/−^ mice, indicating a substantial alteration in muscle fiber composition. This shift from MyoFast to MyoSlow fibers in CB1^−/−^ mice implies a dynamic role of CB1 in regulating the equilibrium between these fiber types (Figure 1). Slow-twitch fibers are known for their higher oxidative capacity, while fast-twitch fibers are associated with more rapid glycolytic energy production. The intriguing aspect of this shift becomes apparent when considering the simultaneous increase in oxidative capacity, as evidenced by elevated COX activity. This suggests that the observed transition in fiber types may be part of an adaptive response aimed at enhancing oxidative capacity in the absence of CB1 signaling. A similar change in muscle fiber composition was recently documented by Gonzalez-Mariscal et al. in skeletal muscle-specific CB1^−/−^ mice [24]. They observed that the genetic ablation of CB1 in muscle influences muscle metabolism by decreasing fat accumulation, promoting oxidative phosphorylation, increasing muscle mass, and inducing a shift in muscle composition characterized by an enrichment of oxidative fibers. Ultimately, these changes contribute to enhanced physical performance and an overall improvement in whole-body metabolism. It should be noted that despite our animal model being a global CB1^−/−^ and the model used by Gonzalez-Mariscal et al. lacking only CB1 in muscle tissue, the changes in the composition of muscle fibers are similar. The alterations in fiber composition, accompanied by elevated muscle oxidative capacity (Figure 1) and an increase in mitochondrial respiration coupled with ATP production in isolated myofibers, as demonstrated by Gonzalez-Mariscal et al., highlight the importance of muscle CB1 in coordinating integrated metabolic regulation [24]. In addition, we reveal a simultaneous increase in the levels of miR-499 and miR-208b, members of the myomiR family, and a correlated decrease in their target genes, such as *sox6*, *rod1*, FNIP1, and TCF12 (Figure 1), providing new insights into studying the mechanism of action of the fiber-type switch regulated by CB1. MicroRNAs, crucial regulators of gene expression, emerge as key players in the context of CB1 signaling, suggesting their potential role in coordinating the molecular pathways that regulate muscle fiber types and oxidative metabolism. MiR-499 and miR-208b, located in the intron of myosin genes, play a pivotal role in energy metabolism and remodeling mature muscle into a slow fiber-dominated, oxidative metabolism phenotype by targeting various genes in skeletal muscle [45,46,47]. There is a correlation between miR-499 expression in slow-twitch fibers and both cleavage of mRNA targets and translational repression of the *sox6* and *rod1* genes [47]. *Sox6* and *rod1* exhibited four and two highly conserved miR-499 binding sites in their 3′ UTR among vertebrates, respectively. These genes are known to play a role in cell proliferation and differentiation. It has been demonstrated that high miR-499 expression in slow-twitch fibers of Nile tilapia corresponds with both mRNA target cleavage and translational repression of *sox6* and *rod1* genes [47]. In cardiac and skeletal muscle, miR-208b targets FNIP1 and TCF12 mRNA. It has been known that miR-208b not only stimulates myogenic cell proliferation and inhibition but also remodels mature muscle into a slow fiber-dominated, oxidative metabolism phenotype. Mice deficient in FNIP1 were significantly enriched with highly oxidative skeletal muscle that is more resistant to fatigue than wild-type muscle [48]. On the other hand, TCF12 is a potent pro-differentiation myogenic factor in skeletal muscle. Further investigation is necessary to clarify the specific signaling pathways through which CB1 modulates the expression of these miRNAs and their subsequent effects on muscle physiology.

The transition from fast-twitch to slow-twitch fibers and the associated changes in metabolic characteristics within the muscle fibers are likely linked to the modulation of mitochondrial function [49,50].

We found that the absence of CB1 induces a significant increase in the oxidative capacity of the gastrocnemius, a finding further supported by a significant increase in gel activity of Complex I and Complex IV and a decrease in gel activity of Complex II (Figure 2). However, the unchanged protein levels of OXPHOS complexes indicate that the changes in enzyme activity may be due to post-translational modifications rather than changes in protein expression. The increase in oxidative capacity appears to come at the cost of enhanced oxidative stress, as shown by the significant increase in H_2_O_2_ levels, an indirect marker of enhanced mitochondrial superoxide production [51].

This state of oxidative stress is aggravated by the observed alterations in the antioxidant defense system, highlighting the role of CB1 in maintaining redox balance. Specifically, we observed a significant decrease in the protein levels of two key antioxidant enzymes, SOD-2 and GPX-1, in CB1^−/−^ mice (Figure 4). The reduction in these crucial enzymes could imply a diminished capacity to neutralize ROS, thereby exacerbating oxidative stress. Similar results were reported in the skin of CB1^−/−^ mice by Leal et al. They hypothesized that the increased ROS production and decreased antioxidant defenses in the skin of CB1^−/−^ mice led to accelerated skin aging [52]. Interestingly, the increase in CATALASE, although not statistically significant, implies a selective attempt to maintain a balance in ROS detoxification, although insufficient to counteract the overall increase in oxidative stress. This differential response in antioxidant enzymes suggests a selective disturbance in the antioxidant defense mechanism rather than a uniform downregulation. The increased CATALASE levels may indicate that some compensatory mechanisms might be at play to maintain a balance in the ROS detoxification process, although these mechanisms appear insufficient to counteract the overall increase in oxidative stress, as indicated by the elevated H_2_O_2_ levels. The increased oxidative capacity and concomitant oxidative stress observed in CB1^−/−^ mice reveal a complex interplay between CB1 signaling and mitochondrial dynamics within the gastrocnemius muscle. The shift from fast-twitch to slow-twitch fibers, characterized by a simultaneous elevation in oxidative capacity, suggests a potential compensatory mechanism. Slow-twitch fibers, known for their higher oxidative capacity, may respond to the heightened energy demands by activating mitochondrial biogenesis. In fact, the increase in markers of mitochondrial biogenesis, including PGC1α, TFAM, and NRF1 protein levels, underscores the impact of CB1 absence on mitochondrial dynamics, a process essential for generating new, functional mitochondria (Figure 5). Indeed, the coordination between fiber-type shift and increased mitochondrial biogenesis may indicate an integrated strategy to optimize muscle function [53,54]. Structurally, mitochondrial homeostasis is closely regulated by mitochondrial dynamics, including mitochondrial fission, mitochondrial fusion, and mitophagy [55,56]. Mitochondrial fusion and fission are crucial processes governing the morphology and function of the mitochondrial network. The observed significant increase in OPA1, coupled with unchanged Mfn2 protein levels, points towards a shift in mitochondrial dynamics favoring fusion over fission. These data are further supported by electron microscope evidence showing larger mitochondria in the gastrocnemius of CB1^−/−^ mice (Figure 5). However, this adaptive response appears to come at the cost of enhanced oxidative stress, as evidenced by elevated H_2_O_2_ levels.

The analysis of mitophagy-related protein levels in the gastrocnemius of CB1^−/−^ mice revealed a significant decrease in p62 and a simultaneous increase in LC3B, Parkin, Pink1, and Ambra1. These alterations strongly indicate an upregulation of mitophagy in the absence of CB1 signaling (Figure 5). The interconnected mitochondrial network resulting from increased fusion, which may be a response to heightened oxidative stress, serves several purposes. First, it allows the exchange of mitochondrial contents, including proteins and genetic material, promoting overall mitochondrial function and efficiency. Second, the fused network facilitates the dilution of damaged components across a larger mitochondrial mass, potentially reducing the impact of dysfunction [57,58]. While fusion promotes mitochondrial network integrity and functionality, it also creates a platform for efficient mitophagy. The elongated, interconnected mitochondria formed during fusion are more easily recognized by the mitophagic machinery of the cell [59,60,61]. In conditions of cellular stress or damage, such as increased oxidative stress, as seen in CB1^−/−^ mice, the cell may activate mitophagy to selectively remove dysfunctional mitochondria.

The observed increase in phosphorylated PERK and eIF2α levels in the gastrocnemius of CB1^−/−^ mice indicates the activation of the UPR^ER^ within the endoplasmic reticulum. In conditions of ER stress, the accumulation of unfolded or misfolded proteins triggers the UPR to restore ER homeostasis. Phosphorylation of PERK and eIF2α is a key step in this adaptive response, temporarily reducing protein synthesis to alleviate the ER load [62,63].

The significant increase in the levels of LONP1, CLPP, TRAP1, ATF4, ATF5, and CHOP proteins, particularly the approximately sevenfold increase in TRAP1, in CB1^−/−^ mice, is a noteworthy finding that sheds light on the UPR^mt^ and overall protein quality control within the mitochondria (Figure 6). The increased levels of these proteins suggest an activation or upregulation of the UPR^mt^. This response is typically triggered when the mitochondria face stress, such as increased ROS production or protein misfolding. It aims to restore mitochondrial function by enhancing the capacity to refold or degrade damaged proteins [64,65]. Given the observed increase in oxidative stress markers, the elevated levels of these proteins might be a protective response, mitigating the detrimental effects of oxidative damage on mitochondrial proteins. By ensuring the proper folding and degradation of mitochondrial proteins, these changes likely help in maintaining mitochondrial integrity and function. This is particularly important in the context of increased mitochondrial biogenesis and activity, as observed in the CB1^−/−^ mice. Since we know that mitochondria and the endoplasmic reticulum are physically connected through inter-organelle contact sites known as mitochondrial-associated membranes (MAMs), this intricate crosstalk may be pivotal in orchestrating a coordinated stress response [66,67]. The activation of UPR^mt^ and UPR^ER^ in response to CB1 deficiency indicates a concerted effort by the cell to manage the stress induced by the absence of CB1 signaling. The simultaneous activation of UPR^mt^ and the observed increase in mitochondrial biogenesis suggest a dynamic interplay between CB1 and the mitochondria. The endocannabinoid system, acting through CB1, likely plays a dual role in maintaining mitochondrial homeostasis by influencing both protein quality control mechanisms and the generation of new mitochondria.

Finally, the intricate regulatory network extends to DNA repair mechanisms, with a significant increase in APE-1 and a lack of variation in OGG1, suggesting a concerted effort to maintain genomic integrity in the face of CB1 deficiency. The non-significant increase in POL-γ further hints at a refined modulation of mitochondrial DNA repair pathways (Figure 5).

The findings from our study on CB1 deficiency in the gastrocnemius muscle have broad physiological implications that extend to muscle performance, endurance, and susceptibility to muscle-related diseases. The multifaceted impact of CB1 deficiency, as highlighted by alterations in muscle fiber composition, oxidative capacity, mitochondrial dynamics, and cellular stress responses, suggests several potential consequences for overall muscle physiology.

The shift from fast-twitch to slow-twitch muscle fibers in CB1^−/−^ mice, coupled with an increase in oxidative capacity, implies a dynamic role of CB1 in regulating muscle fiber types. This shift could potentially impact muscle endurance, making CB1^−/−^ muscles more suited for sustained aerobic activities.

The increase in markers of mitochondrial biogenesis and alterations in mitochondrial dynamics suggest a reprogramming of energy metabolism in CB1^−/−^ mice. The enhanced capacity for mitochondrial biogenesis may be an adaptive response to altered muscle fiber composition, potentially optimizing overall muscle function. However, the observed increase in oxidative stress and changes in antioxidant defense mechanisms may compromise mitochondrial function over time, influencing energy production and overall metabolic balance.

The activation of both UPR^mt^ and UPR^ER^ indicates a comprehensive cellular stress response to CB1 deficiency. While these responses aim to maintain cellular homeostasis, sustained activation may have implications for long-term muscle health. Chronic stress responses can contribute to cellular dysfunction and may influence the susceptibility to muscle-related diseases.

In summary, our study not only advances our understanding of CB1’s involvement in mitochondrial dynamics but also pioneers the exploration of its roles in UPR^mt^, UPR^ER^, and miRNA-mediated regulation of muscle fiber composition. These groundbreaking findings are crucial for advancing therapeutic strategies related to muscle-related disorders, offering valuable insights into optimizing muscle function, and providing a foundation for further research in this emerging field. 

## 5. Conclusions

In conclusion, a complex adaptive response is triggered by CB1 deficiency in order to optimize muscle function. This may offer avenues for developing interventions that target mitochondrial health in muscle-related diseases. However, the long-term consequences of this dynamic interplay raise crucial questions about the cellular cost over time. What does this ongoing balance struggle mean for the cell? The sustained activation of adaptive mechanisms suggests a continuous effort to mitigate the impact of oxidative stress. 

Furthermore, while the study makes significant contributions to the understanding of CB1 in muscle biology, its generalizability may be limited by the global knockout model, the focus on a specific muscle, and the predominantly molecular approach. Future research to deepen our understanding and explore potential applications could be aimed at verifying, for instance, the long-term effects of CB1 modulation on muscle health. In addition, given the complexity of muscle physiology, exploring the crosstalk between CB1 signaling and other relevant pathways would be worthwhile. 

## Figures and Tables

**Figure 1 biology-13-00116-f001:**
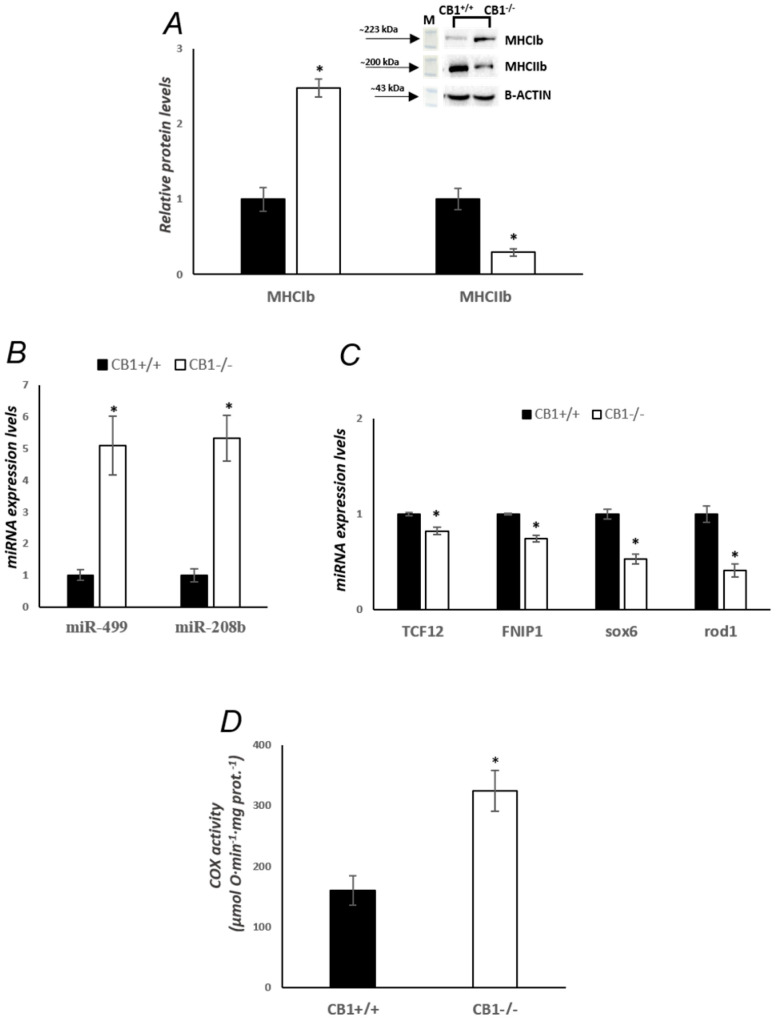
CB1 deletion affects fiber-type switching and oxidative capacity in skeletal muscle. (**A**) Representative immunoblots of MHCIb/B-ACTIN and MHCIIb/B-ACTIN in the skeletal muscle of CB1^+/+^ and CB1^−/−^ mice. Histograms show the results of the densiometric analysis of immunoblots. B-ACTIN was used as a loading control. (**B**) RT-qPCR analysis of miR-499 and miR-208b in the skeletal muscle of CB1^+/+^ and CB1^−/−^ mice. The expression levels of these miRNAs were normalized to a reference gene (RNU6B). (**C**) RT-qPCR analysis of TCF12, FNIP1, *sox6,* and *rod1* in the skeletal muscle of CB1^+/+^ and CB1^−/−^ mice. The mRNA expression levels were normalized to a reference gene (B-ACTIN and GAPDH). (**D**) Activity of cytochrome oxidase in skeletal muscle mitochondria of CB1^+/+^ and CB1^−/−^ mice. Data are represented as mean ± SEM; *n* = 5. * *p* < 0.05 vs. CB1^+/+^ (Student’s *t*-test).

**Figure 2 biology-13-00116-f002:**
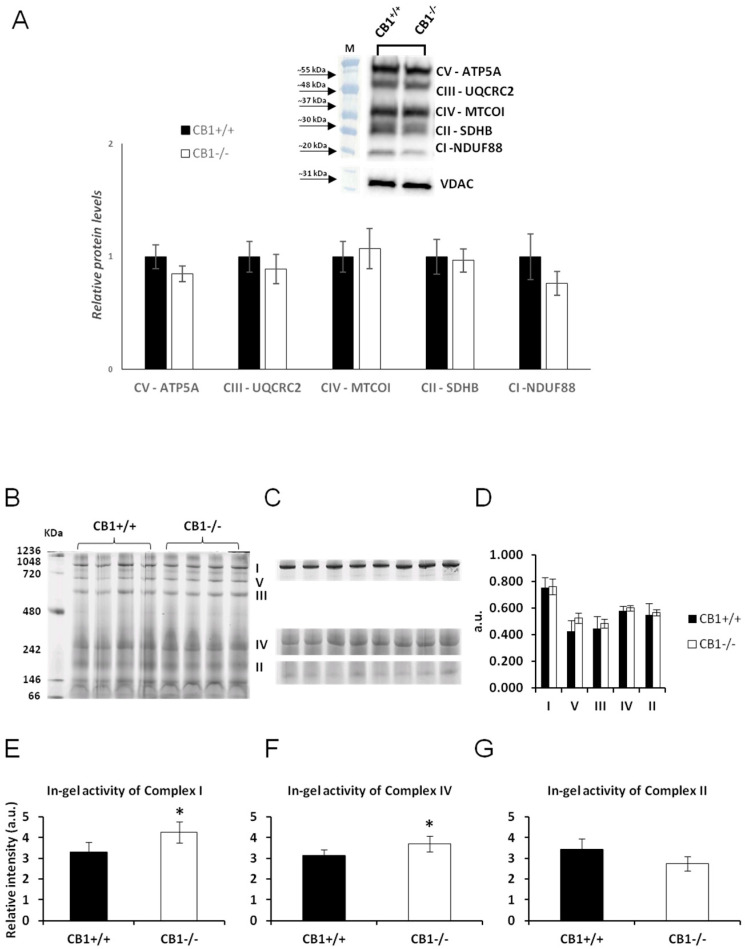
CB1 deletion affects the functional/structural organization of the respiratory chain. (**A**) Representative immunoblots of CI–CV respiratory chain complex protein levels in skeletal muscle mitochondria of CB1^+/+^ and CB1^−/−^ mice. Histograms show the results of the densiometric analysis of immunoblots. B-ACTIN was used as a loading control. (**B**) BN-PAGE-based analysis of individual respiratory complexes from dodecylmaltoside-solubilized crude mitochondria from skeletal muscle of CB1^+/+^ and CB1^−/−^ mice. Representative image of a Coomassie blue-stained BN-PAGE gel. Bands characteristic of individual OXPHOS complexes are recognizable. The molecular weights of standard proteins and the relative position of the respiratory complexes (I–V) are indicated. (**C**) Representative images of histochemical staining of complex I (I), complex IV (IV), and complex II (II) in-gel activity. (**D**) Densitometric quantification of the blue bands corresponding to individual complexes (arbitrary units, a.u.). (**E**–**G**) Densitometric quantification of bands corresponding to individual in-gel activity of complex I, IV, and II (relative intensity, in arbitrary units, a.u.). Protein extracts were prepared for each animal, and each individual was assessed separately. Protein load was 15 µg/lane. Data were reported as relative intensity and presented separately for each treatment (means ± SD; *n* = 5). * *p* < 0.05 vs. CB1^+/+^.

**Figure 3 biology-13-00116-f003:**
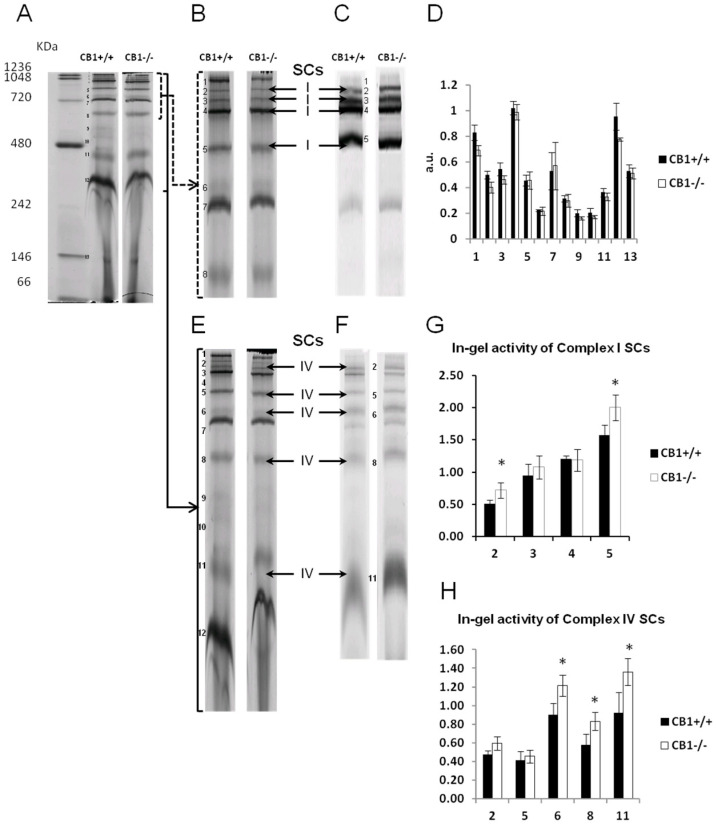
CB1 deletion affects the functional/structural organization of respiratory chain supercomplexes. (**A**) Blue native PAGE-based analysis of digitonin-solubilized crude mitochondria from skeletal muscle of CB1^+/+^ and CB1^−/−^ mice. A representative image of a Coomassie blue-stained BN-PAGE gel. Bands characteristic of OXPHOS supercomplexes (SCs) (bands 1–13) are recognizable in all the experimental groups and highlighted in (**B**,**E**). (**C**,**F**) Representative images of histochemical staining of in-gel activity (arrows) of Complex I and Complex IV SCs. (**D**) Densitometric quantification of the blue bands corresponding to individual SCs. (**G**,**H**) Densitometric quantification of bands corresponding to the in-gel activity of Complex I and IV SCs. Protein extracts were prepared for each animal, and each individual was assessed separately. Protein load was 15 µg/lane. Data were reported as relative intensity and presented separately for each treatment (means ± SD; *n* = 5). * *p* < 0.05 vs. CB1^+/+^.

**Figure 4 biology-13-00116-f004:**
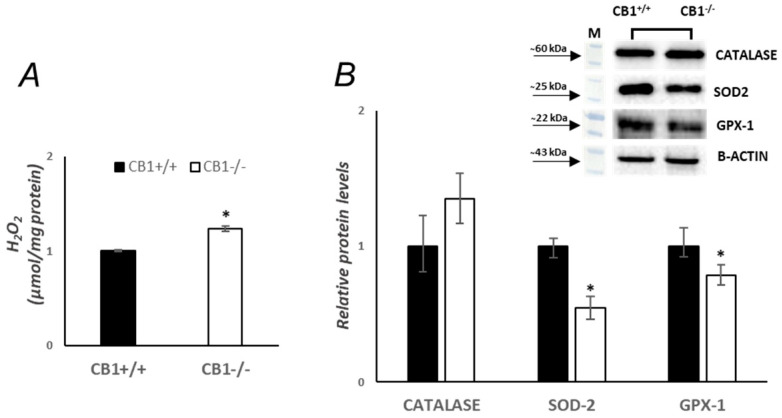
CB1 deletion affects antioxidants’ defense status. (**A**) H_2_O_2_ levels (µmol/mg protein) in the skeletal muscle of CB1^+/+^ and CB1^−/−^ mice. (**B**) Representative immunoblots of CATALASE/B-ACTIN, SOD-2/B-ACTIN, and GPX-1/B-ACTIN in the skeletal muscles of CB1^+/+^ and CB1^−/−^ mice. Histograms show the results of the densiometric analysis of immunoblots. B-ACTIN was used as a loading control. Data are represented as mean ± SEM; *n* = 5. * *p* < 0.05 vs. CB1^+/+^ (Student’s *t*-test).

**Figure 5 biology-13-00116-f005:**
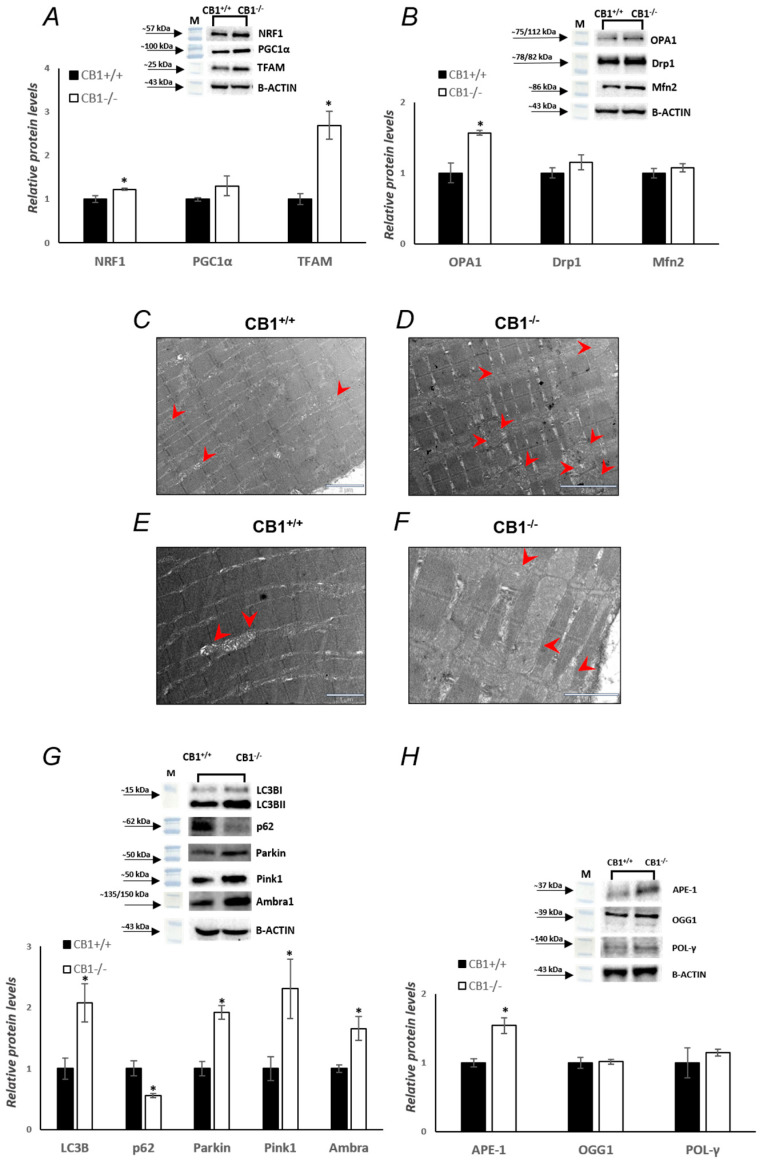
CB1 deletion affects mitochondrial quality control. (**A**,**B**,**G**,**H**) Representative immunoblots of NRF1/B-ACTIN, PGC1α/B-ACTIN, TFAM/B-ACTIN, OPA1/B-ACTIN, Drp1/B-ACTIN, Mfn2/B-ACTIN, APE-1/B-ACTIN, OGG1/B-ACTIN, POL-γ/B-ACTIN, LC3BI/II/B-ACTIN, p62/B-ACTIN, Parkin/B-ACTIN, Pink1/B-ACTIN, and Ambra1/B-ACTIN in skeletal muscle of CB1^+/+^ and CB1^−/−^ mice. Histograms show the results of the densiometric analysis of immunoblots. B-ACTIN was used as a loading control. (**C**–**F**) Transmission electron microscopy analysis of CB1^+/+^ and CB1^−/−^ muscle fibers. Representative images of CB1^+/+^ and CB1^−/−^ muscle fiber ultrastructure. Red arrowheads indicate intermyofibrillar mitochondria. Data are represented as mean ± SEM; *n* = 5. * *p* < 0.05 vs. CB1^+/+^ (Student’s *t*-test).

**Figure 6 biology-13-00116-f006:**
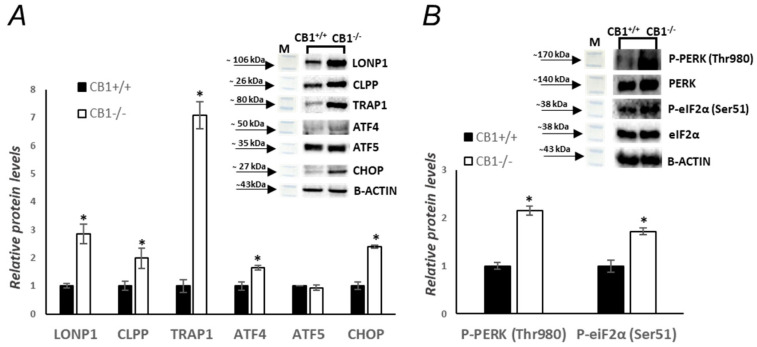
CB1 deletion affects UPR^mt^ and UPR^ER^ processes. (**A**,**B**) Representative immunoblots of LONP1/B-ACTIN, CLPP/B-ACTIN, TRAP1/B-ACTIN, ATF4/B-ACTIN, ATF5/B-ACTIN, CHOP/B-ACTIN, P-PERK (Thr980)/PERK, and P-eiF2α (Ser51)/eiF2α, in skeletal muscle of CB1^+/+^ and CB1^−/−^ mice. Histograms show the results of the densiometric analysis of immunoblots. B-ACTIN was used as a loading control. Data are represented as mean ± SEM; *n* = 5. * *p* < 0.05 vs. CB1^+/+^ (Student’s *t*-test).

**Table 1 biology-13-00116-t001:** The primers used in the study.

Primer	Forward	Reverse
TCF12	5′-GACCAACTACACTGGGAAGCA-3′	5′-GGAAGGACTTGGTTGACCACT-3′
FNIP1	5′-CTGCTCAGAGATGCAGAACG-3′	5′-AATGGACATGCCAGGAAGAG-3′
rod1	5′-AGACCTGCTGCTTGAGGAAA-3′	5′-GGTGCACCGGGTATAATGTC-3′
sox6	5′-ATGCTGCCAGCTTTTTCTGT-3′	5′-GGCAACTCTCCACCATGATT-3′
B-ACTIN	5’-CAACGGCTCCGGCATGTGC-3’	5’-CTCTTGCTCTGGGCCTCG-3’
GAPDH	5’-GTCGTGGATCTAACGTGCC-3’	5’-GATGCCTGCTTCACCACC-3’

## Data Availability

The data presented in this study are available in the article.

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
