# Peer review of "Effect of CB1 Receptor Deficiency on Mitochondrial Quality Control Pathways in Gastrocnemius Muscle"

_biology, 2024, doi:10.3390/biology13020116_

Round 1

Reviewer 1 Report

Comments and Suggestions for Authors

This manuscript, titled "Effect of CB1 Receptor Deficiency on Mitochondrial Quality Control Pathways in Gastrocnemius Muscle," presents a focused study on the role of CB1 receptors in muscle physiology, particularly emphasizing mitochondrial dynamics in skeletal muscle. Key findings include a shift from fast-twitch to slow-twitch muscle fibers in CB1-/- mice, implying CB1's role in muscle fiber type regulation and energy metabolism.

Significantly, the study uncovers complex changes in oxidative stress response and antioxidant defense mechanisms, highlighting CB1's involvement in redox homeostasis. Notably, an increase in mitochondrial hydrogen peroxide production and altered levels of antioxidant enzymes like SOD-2, GPX-1, and CATALASE are observed in CB1-/- mice.

The research delves into mitochondrial quality control, showing enhanced mitochondrial biogenesis and dynamics favoring fusion in the absence of CB1. This is supported by increased levels of proteins related to mitochondrial fusion and mitophagy. Furthermore, the activation of stress responses in both mitochondria (UPRmt) and the endoplasmic reticulum (UPRER) in CB1-/- mice underscores the wider impact of CB1 deficiency on cellular homeostasis.

Overall, the manuscript sheds light on the multifaceted role of CB1 in skeletal muscle, linking it to mitochondrial function and stress responses, and opening potential avenues for future research in muscle health and disease.

To improve the overall quality of your manuscript, the following suggestions are recommended:

Introduction

1. The section is well-structured and provides a detailed overview of the endocannabinoid system, particularly focusing on the cannabinoid type 1 receptor (CB1) and its role in skeletal muscle and mitochondrial function. The flow from general information about endocannabinoids to specific details about CB1 in skeletal muscle is logical and easy to follow.

2. The manuscript successfully integrates a wide range of studies to support its assertions, which strengthens the credibility of the information presented. However, it would be beneficial to include more recent studies or reviews to ensure the most current understanding of CB1's role in skeletal muscle and mitochondrial function is presented.

3. The manuscript makes several broad statements about the role of CB1 in muscle tissue and mitochondrial function. While these are supported by references, more specific examples or data from these studies would enhance the manuscript's persuasiveness and provide clearer insights into the mechanisms involved.

4. The manuscript acknowledges ongoing research and unanswered questions, particularly in the area of muscle fiber-type switching and miRNA involvement in CB1-induced transcriptional regulation. It could benefit from a more detailed discussion of these gaps and potential directions for future research.

5. The technical language and terminology are appropriate for the target audience. However, it might be helpful to briefly explain some of the more specialized terms (e.g., UPRmt, MQC) for readers who may not be as familiar with this specific field.

6. The manuscript could be strengthened by a section discussing the practical implications of these findings. How might this knowledge impact clinical approaches or therapeutic strategies related to muscle-related diseases or conditions?

Results

7. Consider elaborating on the controls used in your experiments. For instance, in Western blot and RT-qPCR analyses, it's crucial to mention the housekeeping genes or proteins used as loading controls in the figure legend. This information is vital for assessing the reliability of your data.

8. The manuscript mentions that each animal was assessed separately, which is good for reproducibility. However, a sample size of n=5 might be considered small for drawing robust conclusions. Discuss the possibility of increasing the sample size to enhance the statistical power of your findings.

9. For Western blot and BN-PAGE analyses, ensure that the densitometric quantification method is thoroughly described. Details like software used, settings, and how background correction was performed can significantly impact the interpretation of your results.

10. Include more detailed methodological information. For example, describe the conditions under which Western blotting and BN-PAGE were performed (e.g., gel concentrations, buffer systems, and electrophoresis conditions). This level of detail is crucial for other researchers attempting to replicate your study.

11. Address the validation of the techniques used. For instance, how was the specificity of antibodies in Western blots confirmed? Were there any controls to confirm the absence of cross-reactivity?

12. While the results indicate changes in the expression of MHC isoforms and myomiRs in CB1-/- mice, it would be beneficial to discuss how these changes correlate with functional outcomes in muscle physiology. Are there any functional assays that could be performed to directly assess the impact of these molecular changes on muscle function?

13. It's mentioned that Student's t-test was used for statistical analysis. However, consider whether this is the most appropriate test, especially if data distribution and variance assumptions are not met. Additionally, provide details on how outliers were handled and whether any data normalization was performed.

14. Compare your results with existing literature more explicitly. How do your findings align with or contradict previous studies on CB1 in muscle physiology? This comparison can add depth to your results and provide a broader context for the reader.

15. The manuscript could benefit from a deeper exploration of the underlying mechanisms. For instance, how might the changes in mitochondrial enzyme activities contribute to the overall muscle phenotype observed in CB1-/- mice? Are there additional experiments that could be conducted to unravel these mechanisms further?

16. The comparison between CB1-/- and CB1+/+ mice is central to your study. It's essential to ensure that the age, sex, and environmental conditions of the animals are matched as closely as possible to rule out confounding factors.

17. For measurements of H2O2, SOD-2, GPX-1, and CATALASE levels, provide specific details about the assays used, including kit brands, concentrations, and reaction conditions. This information is crucial for reproducibility and validity of the results.

18. Consider using more robust statistical analyses beyond the Student's t-test, especially if the data distribution is not normal or if the variances are unequal. Techniques like ANOVA followed by post-hoc tests could offer more insight, especially with small sample sizes.

19. In the transmission electron microscopy analysis, it's crucial to describe the sample preparation process in detail. This includes fixation, sectioning, and staining methods. Ensure that the images are of high resolution and appropriately scaled.

20. When discussing densiometric analysis, specify the software used and how the background was accounted for. For electron microscopy images, ensure that the scale bars are clear and the red arrowheads indicating intermyofibrillar mitochondria are easily identifiable.

21. While discussing the changes in antioxidant defense mechanisms, delve deeper into the biological implications of these changes. How do these alterations in enzyme levels contribute to overall muscle physiology and pathology in the context of CB1 deficiency?

22. The increase in mitophagy markers and mitochondrial fusion proteins should be interpreted in the context of overall mitochondrial health. Discuss whether these changes represent a compensatory mechanism in response to stress or a sign of altered mitochondrial dynamics.

23. The involvement of UPRmt and UPRER pathways is intriguing. Offer more context on how these pathways interplay and what their activation means in terms of cellular stress responses in CB1-/- mice. Additionally, explaining the significance of phosphorylation levels of P-PERK and P-eiF2α in lay terms would be beneficial.

Discussion

24. While you reference the work of Gonzalez-Mariscal et al. and others, more detailed comparisons with these studies would strengthen your discussion. Specifically, how do your findings align or differ from these studies? This comparison will provide a richer context for your results.

25. The increase in miR-499 and miR-208b levels is an interesting finding. Delve deeper into how these miRNAs might mediate the effects of CB1 signaling on muscle fiber type and oxidative metabolism. Discuss potential target genes of these miRNAs and their relevance in muscle physiology.

26. The increased oxidative capacity and oxidative stress in CB1-/- mice are key findings. Discuss the potential mechanisms leading to this state. For instance, is the increased oxidative capacity a compensatory mechanism in response to heightened energy demands of slow-twitch fibers?

27. The differential expression of antioxidant enzymes (SOD-2, GPX-1, and CATALASE) deserves more exploration. Discuss how these changes might reflect a complex regulatory network affected by CB1 signaling. The non-significant increase in CATALASE, in particular, warrants further discussion.

28. Your observations on mitochondrial dynamics and biogenesis are compelling. Discuss how these changes might be related to the shift in muscle fiber types and overall muscle function. How do these findings contribute to our understanding of muscle adaptation in the absence of CB1?

29. The activation of UPRmt and UPRER pathways is an important observation. Elaborate on how these pathways might be interconnected and their implications in maintaining cellular homeostasis under CB1 deficiency.

30. The changes in mitochondrial DNA repair enzymes, such as APE-1 and POL-γ, are intriguing. Discuss the implications of these alterations for mitochondrial integrity and function.

31. Explore the broader physiological implications of your findings. How might CB1 deficiency affect muscle performance, endurance, and susceptibility to muscle-related diseases?

32. Given the role of CB1 in regulating muscle physiology, discuss the potential therapeutic implications of modulating CB1 activity in muscle-related disorders.

33. Suggest specific future research directions based on your findings. For instance, investigating the direct impact of CB1 modulation on muscle function in vivo or exploring the therapeutic potential of targeting CB1 in muscle disorders would be valuable.

Conclusions

34. While you have briefly mentioned the impact of CB1 on muscle fiber composition, oxidative stress response, and mitochondrial dynamics, a slightly more detailed summary of these key findings would be beneficial. For instance, mention the shift from fast-twitch to slow-twitch muscle fibers in CB1-/- mice, the increase in oxidative stress markers, and the alterations in mitochondrial biogenesis and dynamics.

35. Emphasize the novelty and significance of your findings. If your study is one of the first to explore the role of CB1 in mitochondrial dynamics within muscle cells, highlight this. Explain how your findings contribute to the existing body of knowledge and why they are important.

36. Discuss the broader implications of your findings. How do they contribute to our understanding of muscle physiology and pathology? For example, could these insights have implications for conditions characterized by altered muscle fiber composition or mitochondrial dysfunction?

37. Clearly state the potential avenues for future research. What questions remain unanswered? For instance, further investigation might be needed to understand the precise molecular mechanisms through which CB1 regulates mitochondrial dynamics or to explore potential therapeutic applications of modulating CB1 activity in muscle disorders.

38. If applicable, briefly discuss the clinical or therapeutic relevance of your findings. For example, could targeting CB1 pathways offer new strategies for treating muscle-related diseases?

39. It might be beneficial to acknowledge any limitations of your study in the conclusion. This could be related to the experimental models used, the scope of the study, or any other relevant aspects.

Comments on the Quality of English Language

The English language quality of the manuscript appears to be of a high standard. The text is clear, concise, and effectively communicates the scientific findings. The use of technical terminology is appropriate for the intended academic audience, suggesting familiarity with the subject matter and adherence to scientific writing conventions. 

Ensure consistency in the use of specific terms throughout the manuscript. For instance, consistently use either "CB1-/- mice" or "CB1-deficient mice" to avoid potential confusion.

Author Response

We thank the reviewer for the valuable critique and comments, which has contributed to an improved version of the manuscript.

Reviewer 2 Report

Comments and Suggestions for Authors

In a manuscript submitted for review, the Author described the effect of CB1 receptor deficiency on mitochondrial quality control pathways in gastrocnemius muscle.

The work is written correctly and contains all the required parts. However, it requires some corrections and responses to questions and comments.

My questions and comments:

- In the introduction, in the hypothesis of the work, the authors write about the results obtained. This information should not be included in this part of the work, only as a conclusion or in the discussion, please remove it. „The obtained results suggest that CB1plays a crucial role in maintaining mitochondrial integrity and efficiency in skeletal muscle, potentially influencing muscle adaptability and energy metabolism.”

- How many animals were used in the experiment?

- Why were only male mice used in the experiment?

- Please describe the detailed procedure for collecting and dissecting the skeletal muscle, gastrocnemius. Please describe the anatomical structure of the mice’s muscle and what parts it has. Why was this muscle selected for study?

- In conclusion, it would be important to know in which fields of science (medicine?) the results obtained by the Authors could be applied.

Author Response

(The authors gave the same response as above.)

Round 2

Reviewer 1 Report

Comments and Suggestions for Authors

I wish to express my sincere gratitude to the authors for their detailed and careful revisions. The manuscript has shown considerable improvement from its initial submission, reflecting the authors' genuine commitment to resolving the concerns previously outlined.

Comments on the Quality of English Language

Minor editing of English language required